

# bestDEG: a web-based application automatically combines various tools to precisely predict differentially expressed genes (DEGs) from RNA-Seq data

Unitsa Sangket[1,2], Prasert Yodsawat[1], Jiratchaya Nuanpirom[1] and Ponsit Sathapondecha[1,2]

[1] Division of Biological Science, Faculty of Science, Prince of Songkla University, Hat Yai, Songkhla, Thailand
[2] Center for Genomics and Bioinformatics Research, Faculty of Science, Prince of Songkla University, Hat Yai, Songkhla, Thailand

Corresponding author
Unitsa Sangket, unitsa.s@psu.ac.th

## ABSTRACT

**Background.** Differential gene expression analysis using RNA sequencing technology (RNA-Seq) has become the most popular technique in transcriptome research. Although many R packages have been developed to analyze differentially expressed genes (DEGs), several evaluations have shown that no single DEG analysis method outperforms all others. The validity of DEG identification could be increased by using multiple methods and producing the consensus results. However, DEG analysis methods are complex and most of them require prior knowledge of a programming language or command-line shell. Users who do not have this knowledge need to invest time and effort to acquire it.

**Methods.** We developed a novel web application called "bestDEG" to automatically analyze DEGs with different tools and compare the results. A differential expression (DE) analysis pipeline was created combining the edgeR, DESeq2, NOISeq, and EBSeq packages; selected because they use different statistical methods to identify DEGs. bestDEG was evaluated on human datasets from the MicroArray Quality Control (MAQC) project.

**Results.** The performance of the bestDEG web application with the human datasets showed excellent results, and the consensus method outperformed the other DE analysis methods in terms of precision (94.71%) and specificity (97.01%). bestDEG is a rapid and efficient tool to analyze DEGs. With bestDEG, users can select DE analysis methods and parameters in the user-friendly web interface. bestDEG also provides a Venn diagram and a table of results. Moreover, the consensus method of this tool can maximize the precision or minimize the false discovery rate (FDR), which reduces the cost of gene expression validation by minimizing wet-lab experiments.

# INTRODUCTION

The transcriptome is the set of all RNA transcripts in a cell (*Wang, Gerstein & Snyder, 2009*). Understanding the transcriptome is essential for interpreting gene structure,

expression, and regulation, as well as for understanding development and disease. RNA sequencing technology (RNA-Seq), a powerful high-throughput DNA sequencing method, has provided millions of short sequence reads for mapping and quantifying transcriptomes. Differential gene expression analysis using RNA-Seq has become the most popular technique in transcriptome research. Analysis of differentially expressed genes (DEGs) is used in many research areas. In the health and life sciences, analysis of DEGs has been used in identifying cancer biomarkers related to clinical manifestations of the disease (*Parvathareddy et al., 2021*), in identifying genes potentially responsive to drought in crops (*Li et al., 2022*), and in identifying genes responsible for reproduction in economically important animals (*Thepsuwan et al., 2021*; *Wang et al., 2022*). Differential gene expression analysis involves applying statistical tests to gene/transcript expression data and evaluating significant differences in gene expression between two or more experimental conditions. The three main steps in identifying DEGs are read mapping, statistical treatment, and DEG searching. Before read mapping can be started, the RNA-Seq dataset must be validated (*Yodsawat et al., 2021*). RNA-seq expression data can be analyzed by a number of different statistical procedures. The program edgeR (*Robinson, McCarthy & Smyth, 2010*) uses a negative binomial procedure, EB-Seq (*Leng et al., 2013*) uses an empirical Bayesian procedure, NOISeq (*Tarazona et al., 2011*) deploys a nonparametric test, and DESeq2 (*Love, Huber & Anders, 2014*) uses a combination of statistical procedures to first estimate mean–variance dependence and then applies a negative binomial procedure to reduce "noise" in the expression data.

Most tools for differential gene expression analysis are available as R packages. Users can follow the instructions to reproduce or analyze their own data. However, several evaluations of RNA-seq data analysis methods have shown that there is no single method that outperforms all others (*Costa-Silva, Domingues & Lopes, 2017*; *Nguyen et al., 2018*; *Rotllant et al., 2018*; *Waardenberg & Field, 2019*), and each method (program) commonly produces some false positive results. Also, the number of false positive predictions from the current gene expression analysis methods adds costs for validation by wet-lab experiments. An alternative strategy is to use multiple methods and generate—the consensus results to increase the accuracy of DEG identification, but this strategy presents its own difficulty. RNA-seq analysis methods are complex and most of them require knowledge of a programming language or command line shell. Users who do not have this knowledge must invest time and effort to acquire the knowledge for more than one program. In response, software developers and bioinformaticians have attempted to develop more user-friendly software. The most suitable strategy has proved to be the development of a web application that hosts the analysis software.

In this article, we present the bestDEG web application, a new R Shiny web application that identifies DEGs. Shiny is an R-based framework for web application developers. The bestDEG web application analyses the data by several methods and automatically compares the results of each method to improve the accuracy of the predictions.

## MATERIALS & METHODS

### The development of the differential expression analysis pipeline

Most of the tools used for differential expression (DE) analysis have been developed in the R language. In this research, four existing DE analysis packages were used to construct a DE analysis pipeline: edgeR, DESeq2, NOISeq, and EBSeq. These packages were selected because they use different statistical methods to identify DEGs in sequencing data. edgeR uses the canonical negative binomial distribution; DESeq2 estimates the variance-mean dependence before treating the count data with the negative binomial distribution; NOISeq identifies DEGs based on a nonparametric test; and EBSeq uses an empirical Bayesian approach to model features observed in RNA-Seq data.

The input for the pipeline development was the dataset from the pasilla package (http://bioconductor.org/packages/pasilla/). The dataset was compiled from a study of the effects of RNAi knockdown of the splicing factor *pasilla* on a *Drosophila melanogaster* cell culture (*Brooks et al., 2011*). The details of the dataset generation can be found in the pasilla package document. The dataset consists of a read counts table and a sample information table. The read counts table was generated from Raw RNA-Seq reads, which must be previously checked for quality control (*Yodsawat et al., 2021*). The read counts table contains the transcript names in the first column. The following columns contain the total number of reads assigned to these transcripts in different samples. The sample name should start with the experimental condition followed by the number of replicates. An example of a read counts table from the pasilla dataset can be found in Table 1. The sample information table lists the name of each sample and the associated experimental condition. The first column stores the names of the samples with the replicate number. The sample names of the sample information table must match the sample names of the read counts table and be listed in the same order. The name of the experimental condition appears in the second column. This column can identify only two values and must be headed "condition". An example of a sample information table from the pasilla dataset can be found in Table 2.

The script for obtaining the dataset was taken from the DESeq2 package documentation. The dataset was exported in two tab-separated values (TSV) files. The read counts table file was named "cts.tsv" and the sample information table file was named "coldata.tsv". The R script to generate and export a dataset can be found on GitHub (https://github.com/unitsa-sangket/bestDEG).

The bestDEG pipeline is the main analysis script in the Shiny web application. Before creating the bestDEG pipeline script, all DEG pipelines adapted from its documentation were converted to R functions and saved to a file. The function file is then later used in the bestDEG pipeline by the source function. To create the R function, the global R variables were used as function arguments. The pipeline script for DEG analysis (without the global R variable code) was then inserted into the function body. The return function is used to return the DEG analysis results to the function and was placed at the end of the function body. The last variable in each DEG analysis pipeline is used as the return variable. All DEG analysis function files were stored in the directory

**Table 1   The first 10 lines of the read counts table from the dataset of the pasilla package.**

| Transcript_name | Treated1 | Treated2 | Treated3 | Untreated1 | Untreated2 | Untreated3 | Untreated4 |
|---|---|---|---|---|---|---|---|
| FBgn0000003 | 0 | 0 | 1 | 0 | 0 | 0 | 0 |
| FBgn0000008 | 140 | 88 | 70 | 92 | 161 | 76 | 70 |
| FBgn0000014 | 4 | 0 | 0 | 5 | 1 | 0 | 0 |
| FBgn0000015 | 1 | 0 | 0 | 0 | 2 | 1 | 2 |
| FBgn0000017 | 6205 | 3072 | 3334 | 4664 | 8714 | 3564 | 3150 |
| FBgn0000018 | 722 | 299 | 308 | 583 | 761 | 245 | 310 |
| FBgn0000022 | 0 | 0 | 0 | 0 | 1 | 0 | 0 |
| FBgn0000024 | 10 | 7 | 5 | 10 | 11 | 3 | 3 |
| FBgn0000028 | 0 | 1 | 1 | 0 | 1 | 0 | 0 |
| FBgn0000032 | 1698 | 696 | 757 | 1446 | 1713 | 615 | 672 |

**Table 2   The sample information table of the dataset from the pasilla package shown in Table 1.**

| sample_name | condition |
|---|---|
| treated1 | treated |
| treated2 | treated |
| treated3 | treated |
| untreated1 | untreated |
| untreated2 | untreated |
| untreated3 | untreated |
| untreated4 | untreated |

named "R". The R script of the DEG analysis functions was made available on GitHub (https://github.com/unitsa-sangket/bestDEG).

An overview of the bestDEG pipeline process is presented in Fig. 1. To create the bestDEG pipeline script, the global R variables, read counts, sample information, and the DEG functions are defined at the beginning of the script. A global R variable is stored in the list called input. The read counts and sample information tables are stored in the list named internal. Each list is named to match the variable used in the server-side script of the Shiny web application. The R functions used to analyze DEGs are called from the file, using the function source along with the local parameter to avoid a conflict environment. Next, the order of the columns in the read counts table is changed to match the order of the row names in the sample information table. Then, the desired differential expression analysis methods are checked. If a DEG method is checked, the DEG function of that package is used. If not, the DEG function is skipped and another DEG function is searched. Next, the DEG regulatory status column is added to all DEG results. The newly added column contains three values: up, down, and zero_value. "Up" means that the log2 fold change is greater than zero. "Down" means that the log2 fold change is less than zero. "Zero_value" means that the log2 fold change is equal to zero. The consensus method generated intersection results of multiple method selected by a user. Then, the set of intersection genes (or consensus genes) is determined using the Venn function from the gplots package. The final intersection table is created to summarize the bestDEG result in a

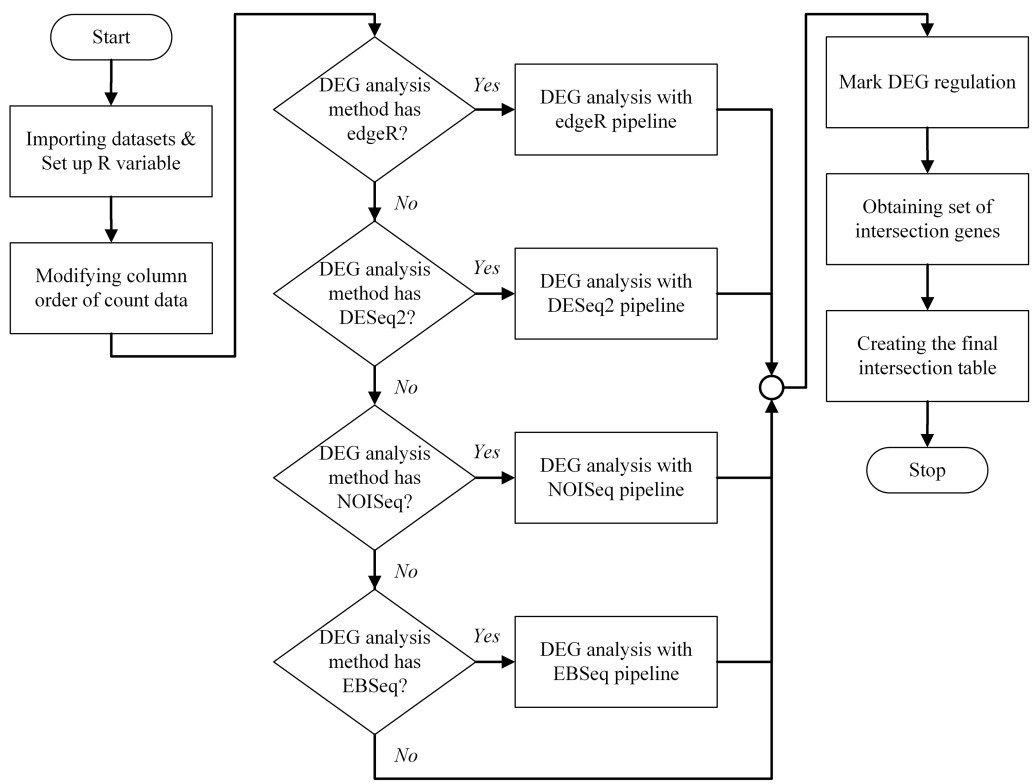

**Figure 1  The program flowchart illustrates the process of the bestDEG pipeline.**

table format. To create the final table, log2 fold change, FDR, and DEG regulatory status are selected from each DEG result using a list of intersection genes. The names of intersection genes are added in the first column. If a gene had an inconsistent DEG regulatory status in the DEG methods, the regulatory status of that gene is marked as "Inconsistent", and the DEG regulatory status is appended to the final table. Next, the log2 fold change from all DEG methods is used to calculate the mean and standard deviation, and then added to the final table. Finally, the FDR and probability value from each DEG method are appended to the last column. The final intersection table displays only those genes whose absolute value of log2 fold change was greater than the threshold (The log2 fold change >= cut-off signified up-regulation while log2 fold change <= -(cut-off) showed down-regulation.) and whose FDR value was less than the threshold. The R script of the bestDEG pipeline is available on GitHub.

## The implementation of the web application

The bestDEG web application was developed in the R programming language and the Shiny package was used to create the web application. The RStudio software was the integrated development environment (IDE) used during development. The renv package was used to manage the package environment. The R code that automatically installs the required R package was stored in the "required_package_install.R" file. The code for the bestDEG

 

web application was stored in several files. The file named "ui.R" stored the UI code that handles all the UI elements of the application. The file named "server.R" was used to store the server code that controls various behaviors, functions, and pipelines of the application. The bestDEG pipeline was embedded in the server code. The R functions for each DE analysis method were stored in the directory named "R". In addition, the file "global.R" was created to specify the global object that will be loaded into the global environment when the application is started. The R script of the bestDEG web application has been published on the bestDEG GitHub, and the bestDEG web application is freely available at: https://unitsa.shinyapps.io/bestDEG.

## Evaluation of the web application

To evaluate the performance of the bestDEG web application, we used two datasets from a previous study (*Costa-Silva, Domingues & Lopes, 2017*) that had been taken from the MicroArray Quality Control (MAQC) project (*Canales et al., 2006*; *Shi et al., 2006*). Two types of RNA sample were used in this project: the Human Brain Reference (HBR) (Ambion; Thermo Fisher Scientific, Waltham, MA, USA) and the Universal Human Reference (UHR) (Stratagene; Agilent Technologies, Santa Clara, CA, USA). Samples were analyzed using RNA sequencing and qRT-PCR. A next-generation sequencing dataset was generated using the Genome Analyzer II sequencing system (Illumina, San Diego, CA, USA). The qRT-PCR dataset was analyzed using TaqMan Gene Expression Assays (Applied Biosystems; Thermo FisherScientific, Waltham, MA, USA). The qRT-PCR dataset was used as the gold standard for evaluating the results of the DEG analysis.

To prepare the RNA sequencing datasets for analysis, Bioconda software (*Grüning et al., 2018*) was used to manage the software environment and installation. Raw datasets were retrieved from the Sequence Read Archive (SRA) database on the National Center for Biotechnology Information (NCBI) website under accession number SRA010153. Samples that used the PhiX control were selected. The accession list for each sample was used to retrieve raw data in FASTQ format using the SRA Toolkit software (version 2.10.9) (https://github.com/ncbi/sra-tools). The human reference genome (GRCh38.p13) and gene annotation in GTF format were downloaded from the Ensembl database (Ensembl Release 102 - November 2020) (http://nov2020.archive.ensembl.org/Homo_sapiens/Info/Index). Prior to further analysis, raw sequencing data were quality checked with the FastQC software (version 0.11.9) (https://www.bioinformatics.babraham.ac.uk/projects/fastqc) and then merged into a single HTML file with the MultiQC software (version 1.9) (*Ewels et al., 2016*). STAR (version 2.7.7a) (*Dobin et al., 2013*) was used to map reads to the human reference genome, and the HTSeq software (version 0.13.5) was used to count the number of reads that mapped to each gene. The read counts of each sample (HBR and UHR) were merged into a single file using an in-house R script, and the sample information file was also created using an in-house R script. Finally, the read counts and sample information files were input into the bestDEG web application for analysis with default settings.

To prepare the qRT-PCR dataset for analysis with an in-house R script, the qRT-PCR data were retrieved from the Gene Expression Omnibus (GEO) database under accession number GSE5350 and platform GPL4097. The dataset was analyzed for DEGs using

the GEO2R Web tool (https://www.ncbi.nlm.nih.gov/geo/geo2r/). The UHR sample was considered a control and DEG analysis was performed using default parameters. Next, the gene symbol identifiers from the GEO2R result were converted to Ensembl Gene ID using db2db in the bioDBnet web tool (*Mudunuri et al., 2009*) with default settings. Before uploading the data to bioDBnet, records with empty, redundant, and invalid gene symbol formats were removed and extracted into a list of gene symbols. After we obtained the result from bioDBnet, gene ID that contained a hyphen symbol or multiple records was removed. The remaining Ensembl gene ID was merged with the previous GEO2R result. Records that did not contain the Ensembl Gene ID were filtered out. Finally, DE was identified by applying the log2 fold change and *p*-value threshold to the GEO2R result. If the gene expression was more than or equal to the absolute value of 2 and the *p*-value was less than or equal to 0.05, the gene was considered a DEG. The datasets marked DEG were labeled "DEG" and "Up" or "Down" in the last columns.

To assess performance, the human RNA-Seq dataset (HBR and UHR) was analyzed using the bestDEG web application. The DEGs were identified based on absolute log2FC $>2$ for upregulated genes and log2FC $<-2$ for downregulated genes with FDR $<0.01$ (*Zheng et al., 2020*; *Teng et al., 2020*; *Methot et al., 2021*; *Vignesh et al., 2021*). The obtained consensus result and the result obtained from each DE analysis package were used to evaluate the performance statistics. The DEG results from the qRT-PCR dataset were used as the gold standard for comparison with the DEG results from the human RNA-Seq dataset. The performance statistics of the consensus method and individual DE analysis methods were calculated using the R package "caret" and compared. To calculate the performance statistics, a confusion matrix was created, which was a table with two dimensions ("Actual" and "Predicted"), each with two classes ("Positive" and "Negative"). Each value in the confusion matrix indicated the number of true positives (TP), false positives (FP), false negatives (FN), and true negatives (TN).

In this study, the positive class was a gene identified as a DEG, and the negative class was a gene identified as not a DEG. A true positive (TP) was the number of genes identified as DEGs in both the predicted and actual data. False positive (FP) was the number of genes identified as DEGs in the predicted data but not in the actual data. False negative (FN) was the number of genes that were not identified as DEGs in the predicted data but were DEGs in the actual data. True negative (TN) was the number of genes identified as not DEGs in both predicted and actual data.

The performance measurement statistics used in this study were described as follows: sensitivity (also recall or the true positive rate) was the ratio between true positive and the sum of true positive and false negative; specificity (also selectivity or the true negative rate) was the ratio between true negative and the sum of true negative and false positive; and precision (or the positive predicted value) was the ratio between true positive and the sum of true positive and false positive.

## RESULTS

The HTML report of fastqc showed a summary of the modules which were run, and a quick evaluation of the results of the module was entirely normal (green tick)

(https://dnacore.missouri.edu/PDF/FastQC_Manual.pdf). The DEG result of the human RNA-Seq dataset from the bestDEG web application is shown in Fig. 2. The performance assessment result is shown in Table 1. The number of DEGs identified from the bestDEG web application for the human RNA-Seq dataset were 5,188 DEGs for edgeR, 8,661 for DESeq2, 8,009 for NOISeq, 5,363 for EBSeq, and 4,662 for the consensus. For specificity and precision, the consensus method outperformed the other DEG analysis methods. For sensitivity, the DESeq2 method outperformed the other DEG methods.

## DISCUSSION

None of the five methods employed in this analysis outperformed the others (Table 3). However, the consensus method increased the precision and specificity of the results. To reduce the cost of gene expression validation by minimizing wet-lab experiments, the number of correct positive predictions (TP) must be maximized, or the false discovery rate (FDR) minimized. Therefore, precision is the most important value to consider because precision is calculated by dividing the number of correctly predicted DEGs (TP) by the sum of all predicted DEGs (TP + FP).

To summarize the advantages of the unique features of the bestDEG web application, we present a comparison of existing software packages developed for multi-method analysis of DE from RNA-Seq data. iDEP (Ge, Son & Yao, 2018) is a web application developed for DE and pathway analysis from RNA-Seq data. This application was developed, like bestDEG, in Shiny but implements DESeq2, limma-voom, and limma-trend methods for DE analysis. However, there are no options to compare outputs and generate consensus DEG results. IDEAMEX (Jiménez-Jacinto, Sanchez-Flores & Vega-Alvarado, 2019) is another web server tool for integrated RNA-Seq data analysis. Nevertheless, this tool does not specify the up- or down-regulated genes. Users have to specify the up- or down-regulated genes by themselves. Also, there is no consensus result for only up- or down-regulated genes from IDEAMEX. consensusDE (Waardenberg & Field, 2019) is an R package that identifies DEGs by combining results from multiple DE analysis methods and includes an option to integrate RUV to improve DE stability. However, users need prior knowledge of R programming and must install R and the associated R package on their own computer. In comparison, the bestDEG web application was developed as a simple and easy-to-use software to perform DE analysis in three simple steps: (1) upload the read counts and sample information, (2) select the DEG method or methods and parameters, and (3) click the Submit button. Users can access the application through the web browser and do not need to install any software or have any programming knowledge to use the application. The bestDEG application implements four DE analysis tools that use different statistical methods to analyze DE in RNA-Seq data. It then automatically generates a consensus DEG result, a plot, a graph and a table that are displayed on the user interface and users can download all their results from the web application.

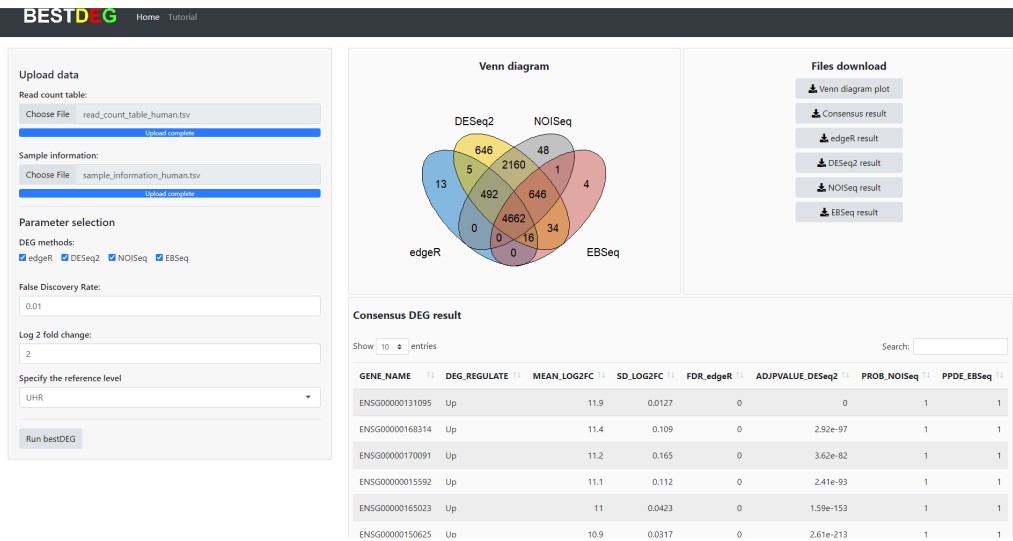

**Figure 2** The DEG results of human RNA-Seq dataset from the bestDEG web application.

**Table 3** The performance of the consensus result from bestDEG was assessed by comparison with the four individual DE analysis methods available in bestDEG.

| DEG method | Sensitivity (%) | Specificity (%) | Precision (%) |
|---|---|---|---|
| Consensus | 70.4188 | **97.0178** | **94.7183** |
| DESeq2 | 86.1256 | 95.6262 | 93.7321 |
| edgeR | 79.8429 | 96.4214 | 94.4272 |
| NOISeq | 84.8167 | 95.2286 | 93.1034 |
| EBSeq | 71.9895 | 96.4214 | 93.8566 |

**Notes.**
*The bold values indicate that the precision and specificity of consensus method are higher than other methods.

# CONCLUSIONS

The bestDEG web application automates the identification of DEGs and optimizes the analysis of differential gene expression in RNA-Seq datasets. bestDEG not only performs differential expression analysis using four methods based on different statistical approaches but also combines the outputs to produce a consensus result. The application allows users to select the desired DE analysis method/s and parameters in the user-friendly web interface. It also provides a Venn diagram and a table of results. In addition, users can download the results of all the DE analysis methods selected and the Venn diagram. The performance of the bestDEG application was evaluated on the gold standard dataset and showed excellent results, especially in the precision and specificity of the consensus method. In addition, the consensus results reported by bestDEG could improve the quality of prediction, which is important for further analysis and validation of gene expression.

# ACKNOWLEDGEMENTS

We thank Mr. Thomas Coyne for language proofreading.

### Funding

This work was supported by the government budget (or budget revenue) of Prince of Songkla University (Grant No. SCI590179S). Prasert Yodsawat was financially supported by the Graduate Fellowship (Research Assistant), Faculty of Science, Prince of Songkla University, Contract no. 1-2561-02-007. The funders had no role in study design, data collection and analysis, decision to publish, or preparation of the manuscript.

### Grant Disclosures

The following grant information was disclosed by the authors:
Prince of Songkla University: SCI590179S.
Faculty of Science, Prince of Songkla University: 1-2561-02-007.

### Competing Interests

The authors declare there are no competing interests.

### Author Contributions

- Unitsa Sangket conceived and designed the experiments, performed the experiments, analyzed the data, prepared figures and/or tables, authored or reviewed drafts of the article, and approved the final draft.
- Prasert Yodsawat conceived and designed the experiments, performed the experiments, analyzed the data, prepared figures and/or tables, authored or reviewed drafts of the article, and approved the final draft.
- Jiratchaya Nuanpirom conceived and designed the experiments, authored or reviewed drafts of the article, and approved the final draft.
- Ponsit Sathapondecha conceived and designed the experiments, authored or reviewed drafts of the article, and approved the final draft.

### Data Availability

bestDEG is available at: https://unitsa.shinyapps.io/bestDEG/.
The source code is available at: https://github.com/unitsa-sangket/bestDEG; unitsa-sangket. (2022). unitsa-sangket/bestDEG: The first release (1.0.0). Zenodo. https://doi.org/10.5281/zenodo.7223314.

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
