# Peer review of "bestDEG: a web-based application automatically combines various tools to precisely predict differentially expressed genes (DEGs) from RNA-Seq data"

_PeerJ, doi:10.7717/peerj.14344_

## Round 0.1 · original submission · Major Revisions

We have received 3 reviews. Please check all the comments demanding revision. Pay attention to the comments by reviewer #2. Main recommendation is show some application, rather than theoretical comparison of the existing tools on an artificial dataset. The journal readers, beginners in bioinformatics need recommendations what to use for differential gene expression analysis. The software should be available.

Reviewer 1 ·

Basic reporting

The quality of English is reasonably good. It is clear and professional. Introduction can be improved. Literature is relevant while the structure is well within the standards of Peer J. Figures are satisfactory.

Experimental design

All the methods that are used are robust are well described. The research question is well defined. Methods are described with sufficient detail and information to replicate.
Authors should highlight the issue of false positive results, which is common in such software.

Validity of the findings

The authors have provided underlying data. Although the study is novel. Its impact remains to be determined. There are multiple similar applications present. However, the fact that it is user-friendly might provide some advantage, which is yet to be determined. It might prove to be a valuable addition to the existing literature.

The conclusion supports the results and effectively summarizes the study. Moreover, the conclusions answer the original research question.

Annotated reviews are not available for download in order to protect the identity of reviewers who chose to remain anonymous.

Reviewer 2 ·

Basic reporting

1. DEG was identified with its raw p-value < 0.05. Multiple comparison method such as Benjamini-Hochberg should be considered.

2. QC (quality check) details were not included.

3. AUROC (Area Under ROC curve) could be a good metric by providing both sensitivity and specificity simultaneously.

4. There is no definition of "consensus" method.

5. R codes to produce the Shinyapp should be shared.

Experimental design

Nothing to add.

Validity of the findings

1. Using bulk-RNAseq data, there are already many benchmarking and reviewing papers in the past 10 years at least. From the systemic review, superiority of DESeq2's performance is not a new information. That is, I am not convinced on what is the novelty and empirical impact of this study.

2. In reality, GS (Golden Standard) data is expensive so that many people rather utilize simulation study. I am questioning on the practical popularity/usefulness of the application for most people who do not have access to GS data.

·

Basic reporting

The article “bestDEG: a web-based application automatically combines various tools to precisely predict diûerentially expressed genes (DEGs) from RNA-Seq data” was submitted by the authors in the journal. In this study, the author has developed a differential expression (DE) analysis pipeline combining the edgeR, DESeq2, NOISeq, and EBSeq packages.
I believe that the manuscript has merit in the current version. However, in my opinion, this work needs minor improvement for further consideration.

Experimental design

-Author should compare the other differential expression (DE) analysis pipeline with their pipeline from example: IDEAmex (PMID: 30984248).

Validity of the findings

-Author should compare the other differential expression (DE) analysis pipeline with their pipeline from example: IDEAmex (PMID: 30984248).
Minor:
Abstract
-The current abstract is lack of a focus point. It should be clear and concise.
Others,
-The author should revise the manuscript for typos and grammatical errors.
Ex. Discussion section: Line 248 “betDEG” should be bestDEG

Additional comments

No commnets

---

## Round 0.2 · accepted · Accept

Thanks for the manuscript update. Two reviewers have no more remarks.

Reviewer 1 ·

Basic reporting

Manuscript is clearly written. Moreover, it has improved substantially after the revision.

Experimental design

It meets the requirements of the paper. methods have been described with adequate detail and can be replicated.

Validity of the findings

Authors have provided all the data. Conclusion addresses the main research question.

Additional comments

In my opinion, after revision, manuscript has been improved. It is suitable for publication.

·

Basic reporting

Clear and unambiguous, professional English used throughout.

Experimental design

Original primary research within Aims and Scope of the journal.

Validity of the findings

Impact and novelty not assessed. Meaningful replication encouraged where rationale & benefit to literature is clearly stated